# HDAC6 Enhances Endoglin Expression through Deacetylation of Transcription Factor SP1, Potentiating BMP9-Induced Angiogenesis

**DOI:** 10.3390/cells13060490

**Published:** 2024-03-11

**Authors:** Chen Sun, Kuifang Xie, Lejie Yang, Shengyang Cai, Mingjie Wang, Yizhun Zhu, Beibei Tao, Yichun Zhu

**Affiliations:** 1Shanghai Key Laboratory of Bioactive Small Molecules, Department of Physiology and Pathophysiology, School of Basic Medical Sciences, Fudan University Shanghai Medical College, Shanghai 200032, China; 16111010002@fudan.edu.cn (C.S.); 20111010003@fudan.edu.cn (K.X.); 22211010004@m.fudan.edu.cn (L.Y.); 16301010003@fudan.edu.cn (S.C.); maggiefd@fudan.edu.cn (M.W.); 2State Key Laboratory of Quality Research in Chinese Medicine and School of Pharmacy, Macau University of Science and Technology, Avenida WaiLong, Taipa, Macau 999078, China; yzzhu@must.edu.mo; 3Department of Pharmacology, School of Pharmacy, Fudan University, Shanghai 200433, China

**Keywords:** histone deacetylase 6, endoglin, acetylation, angiogenesis, endothelial cells

## Abstract

Histone deacetylase 6 (HDAC6) plays a crucial role in the acetylation of non-histone proteins and is notably implicated in angiogenesis, though its underlying mechanisms were previously not fully understood. This study conducted transcriptomic and proteomic analyses on vascular endothelial cells with HDAC6 knockdown, identifying endoglin (ENG) as a key downstream protein regulated by HDAC6. This protein is vital for maintaining vascular integrity and plays a complex role in angiogenesis, particularly in its interaction with bone morphogenetic protein 9 (BMP9). In experiments using human umbilical vein endothelial cells (HUVECs), the pro-angiogenic effects of BMP9 were observed, which diminished following the knockdown of HDAC6 and ENG. Western blot analysis revealed that BMP9 treatment increased SMAD1/5/9 phosphorylation, a process hindered by HDAC6 knockdown, correlating with reduced ENG expression. Mechanistically, our study indicates that HDAC6 modulates ENG transcription by influencing promoter activity, leading to increased acetylation of transcription factor SP1 and consequently altering its transcriptional activity. Additionally, the study delves into the structural role of HDAC6, particularly its CD2 domain, in regulating SP1 acetylation and subsequently ENG expression. In conclusion, the present study underscores the critical function of HDAC6 in modulating SP1 acetylation and ENG expression, thereby significantly affecting BMP9-mediated angiogenesis. This finding highlights the potential of HDAC6 as a therapeutic target in angiogenesis-related processes.

## 1. Introduction

Acetylation stands as one of the most pivotal post-translational modifications (PTMs) applied to both histone and non-histone proteins. Proteins subjected to acetylation often exhibit altered physical and biochemical characteristics, which are essential for cellular adaptation to environmental fluctuations. The balance of protein acetylation within the cell is delicately controlled by two primary enzyme groups: histone acetyltransferases (HATs) and deacetylases (HDACs) [1].

Within the HDAC family, histone deacetylase 6 (HDAC6) emerges as a distinctive member, playing an integral role in modulating a variety of cellular processes, including cell proliferation [2], migration [3,4], stress response [5,6], and endocytosis [7,8] through its deacetylase and non-deacetylase activity. Unique to HDAC6 is its structural composition, featuring two tandem catalytic domains (CD1 and CD2) [9,10]. This dual-domain configuration is an exclusive trait amongst HDACs, endowing HDAC6 with an enhanced and diversified substrate specificity. For instance, DEAD-box helicase 3 X-linked (DDX3X) has been identified as a substrate for both CD1 and CD2 domains [11], while α-tubulin, cortactin, and heat shock protein 90 (HSP 90) are known as specific substrates for the CD2 domain [4,9,10,12,13]. Beyond its deacetylase domains, HDAC6 incorporates a cytoplasmic anchoring domain, predominantly localizing the protein within the cytoplasm, as well as a zinc finger motif (ZNF) for ubiquitin recognition which is involved in protein degradation and autophagy processes [14,15,16]. Notably, HDAC6 is extensively expressed in various systems and cell types, including a crucial involvement in angiogenesis in the cardiovascular system; yet, its molecular mechanisms in this context remain to be fully elucidated [4,17].

Angiogenesis is a vital physiological process that involves the formation of new blood vessels from pre-existing ones. This process is crucial in embryonic development [18], wound healing [19], inflammation response [20], and tumor development [21] and is orchestrated by receptors on the surface of endothelial cells such as vascular endothelial growth factor receptor (VEGFR), which detects circulating vascular endothelial growth factor (VEGF) and initiates signaling pathways that foster the proliferation and migration of endothelial cells (ECs) [22,23,24]. HDAC6 has been reported to influence EC functions via the VEGF pathway [25]. Endoglin (ENG), also known as CD105, is another receptor that has been implicated in angiogenesis [26]. It is a membrane glycoprotein primarily recognized for its role as a part of the transforming growth factor beta (TGF-β) receptor superfamily [27,28] and is predominantly expressed in endothelial cells and functions as a receptor of bone morphogenetic protein 9 (BMP9) [28,29]. Endoglin plays a pivotal role in angiogenesis and in maintaining the structural integrity of the blood vessels; certain mutations of ENG are associated with hereditary hemorrhagic telangiectasia (HHT) [30,31]. However, the regulatory mechanisms of ENG protein in BMP9-induced angiogenesis, and whether ENG is regulated by HDAC6, remain to be explored.

The present study aims to illuminate the connection between HDAC6 and ENG and investigate HDAC6’s role in BMP9-mediated angiogenesis. Furthermore, through transcriptomic and proteomic analyses in endothelial cells, we seek to unravel the intricate mechanisms by which HDAC6 influences this angiogenic process.

## 2. Materials and Methods

### 2.1. Cell Culture

Human umbilical vein endothelial cells (HUVECs) were obtained from Sciencell (San Diego, CA, USA) and cultured according to suppliers’ instructions, with cells being passaged at a 1:3 ratio. Only HUVECs from passages 4 to 6 were utilized for the experiments. Human embryonic kidney 293T cells (HEK 293T) were obtained from ATCC (Manassas, VA, USA) and cultured in DMEM medium (BloomStem, Hainan, China) supplemented with 10% fetal bovine serum at 37 °C with 5% CO_2_.

### 2.2. Plasmid Construction

For HDAC6 knockdown, shRNA sequences 5′ CCTCACTGATCAGGCCATATT 3′ (shHDAC6-1) and 5′ CGGTAATGGAACTCAGCACATC 3′ (shHDAC6-2) were cloned into a modified pLKO.1 vector. For ENG knockdown, shRNA sequence 5′ CCCTGTCATTTGAACCTGGAT 3′ was used, and shRNA sequence 5′ CAACAAGATGAAGAGCACCAA 3′ was used as a scramble control (shNC).

The HDAC6 coding sequences were cloned from a cDNA library derived from HUVECs by PCR. Point mutations were introduced by site-directed mutagenesis. Different constructs were then inserted into a modified pCDH-CMV vector by seamless cloning. The ENG promoter was cloned from the HUVEC genome by PCR and inserted into a pGL3-basic vector by seamless cloning.

### 2.3. Lentivirus-Mediated Gene Knockdown and Overexpression

The lentiviral vectors were co-transferred with other packaging plasmids (pREV, pVSV-G, pTAT, and pGAG) into HEK 293T cells. After 48 h and 72 h, cell medium containing the virus were harvested, and these lentiviruses were purified and quantified as previously described [32]. The HUVECs from passage 4 were infected with a serious amount of lentivirus, cultured for 48 h, and then selected with 2 μg/mL puromycin for another 48 h, after which the cells underwent one more passage before the subsequent experiments.

### 2.4. CRISPR-Cas9-Based HDAC6 Knockout in 293T Cells

Single guide RNA sequence 5′ ACATGATCCGCAAGATGCGC 3′, which targets the human HDAC6 gene, was ligated into a lentiCRISPR v2 vector. HEK 293T cells were transfected with the plasmid. After 48 h, these transfected cells were diluted and cultured for single clone screening. The positive HDAC6 knockout clones were passaged for subsequent experiments.

### 2.5. Wound Healing Assay

The HUVECs were counted and seeded into a 6-well plate and cultured until 100% confluence and then synchronized with ECM basal medium containing 0.5% FBS for 12 h. The monolayer cells were scratched with a plastic pipette; detached cells were washed away 3 times by PBS. Images of the wound were taken at 0 h and 24 h after the scratch and subsequently quantified by ImageJ 1.53e software.

### 2.6. Tube Formation Assay

Growth factor-reduced Matrigel (Sigma-Aldrich, St. Louis, MO, USA) was plated evenly in a 24-well plate and incubated at 37 °C for 60 min before seeding the HUVECs. The HUVECs were pre-synchronized with ECM basal medium containing 0.5% FBS for 12 h. Then, fifty thousand cells were counted and seeded into the solidified Matrigel and incubated for 6 h before photographing. Tube length and branching points were quantified using the ImageJ 1.53e software.

### 2.7. The Transwell Assay

The transwell insert (BD Bioscience, San Jose, CA, USA) was coated with collagen II (ThermoFisher Scientific, Waltham, MA, USA) before seeding cells. Forty-five thousand pre-synchronized HUVECs were seeded into each 24-well transwell insert, and 1 mL ECM basal medium containing 0.5% FBS was added to the lower chamber of the plate and cultured for 24 h. The remaining inside cells were removed by cotton swipe and washed thoroughly with PBS; the migrated cells were fixed with 4% polyformaldehyde, stained with crystal violet, imaged under the microscope, and counted visually.

### 2.8. Immunoprecipitation

Cells were lysed with a binding buffer (50 mM HEPES, 150 mM NaCl, 1% Triton X-100, pH 7.5, supplemented with protease inhibitor cocktail) for 30 min on ice. The cell lysate was cleared by centrifugation at 20,000× *g* for 20 min at 4 °C and quantified by a BCA kit. The cell extracts were then incubated with acetyl-lysine antibody under rotation at 4 °C overnight. The next day, 40 μL of protein A/G agarose beads (ThermoFisher Scientific, Waltham, MA, USA) were added and incubated at room temperature for 1–3 h, and the bound protein was eluted by SDS-PAGE loading buffer heated to 98 °C for 5 min. The HA-tagged proteins were immunoprecipitated with anti-HA affinity agarose beads in a similar way.

### 2.9. The Dual-Luciferase Reporter Assay

The pGL3 plasmid, which contains ENG promoter and pRL-TK plasmid, was co-transfected to HEK 293T cells at a ratio of 100:1. After 48 h, these transfected cells were lysed and cleared with centrifugation. Subsequent luciferase activities were measured under the manufacturers’ instructions in a SpectraMax i3x plate reader (Molecular Devices, Silicon Valley, CA, USA).

### 2.10. Immunofluorescence Staining

The HUVECs were seeded into a 35 mm glass-bottom culture dish, fixed with 4% paraformaldehyde, and permeabilized with 0.1% Triton X-100. HDAC6 and SP1 were stained with corresponding primary antibodies and fluorescent secondary antibodies (Cell Signaling Technology, Beverly, MA, USA), and cell nuclei were stained with DAPI. Confocal laser scanning microscopy was carried out using an LSM710 confocal microscope (Zeiss, Oberkochen, Germany).

### 2.11. SDS-PAGE and Western Blot

Total protein was extracted from the treated HUVECs, and protein concentrations were determined by a BCA protein assay kit (Shenneng Bocai Biotechnology Co., Ltd., Shanghai, China). Equal amounts of protein samples were separated by 10% SDS-PAGE gels and transferred onto 0.45 μm PVDF membranes (Millipore-upstate, Billerica, MA, USA). Membranes were blocked with 5% non-fat milk for 1 h at room temperature and incubated with the primary antibodies overnight at 4 °C. After incubation with the corresponding secondary antibodies, the target protein bands were detected with ImageQuant LAS 4000 (General Electric Co., Fairfield, CT, USA) or Touch imager xli (eBLOT, Shanghai China) and quantified using ImageJ 1.53e software. Detailed antibody information is listed in Appendix A.

### 2.12. High-Throughput Transcriptomic Analysis

The HUVECs were lysed with RNAiso plus reagent (Takara, Kyoto, Japan), and RNA was extracted under the manufacturers’ instruction, among which the mRNA was enriched by oligo magnetic beads and then fragmented. The first strand of the cDNA template was generated by random hexamers; the second strand was synthesized by PCR. The double-strand DNA was purified with the AMpure XP system (Sigma-Aldrich, St. Louis, MO, USA). The cDNA library quality was assessed on an Agilent Bioanalyzer 4150 system (Agilent Technologies, Santa Clara, CA, USA) and sequenced on an Illumina Novaseq 600 (Illumina, San Diego, CA, USA). Differential expression analysis was performed using the DESeq2 [33]; DEGs with |log_2_FC| > 1 and Padj < 0.05 were considered to be significantly different expressed genes.

### 2.13. Reverse Transcription and Real-Time PCR

The RNA was isolated from HUVECs and quantified by OD 260, and then 1 μg RNA was used for reverse transcription by ReverTra Ace™ qPCR RT Master Mix with gDNA remover (Toyobo, Shimahama, Osaka, Japan). Gene amplification was performed by real-time PCR using a SYBR Green Real-Time PCR Kit (Toyobo, Shimahama, Osaka, Japan) on a Light Cycler Pro system (Roche, Basel, Switzerland). The sequence of primers used in the Real-time PCR is listed in Appendix A.

### 2.14. Tandem Mass Tags (TMT) Labeling Proteomic Analysis

The HUVECs were lysed with SDT buffer (4% (*w*/*v*) SDS, 100 mM Tris-HCl pH 7.6, 0.1 M DTT); cell lysate was cleared by centrifugation; and protein concentration was measured by BCA kit. The protein was digested with trypsin by a filter-aided proteome preparation (FASP), and the peptide was quantified by OD280. For each sample, 100 μg of peptide underwent TMT labeling following the manufacturer’s instructions (Thermo Fisher, Waltham, MA, USA). The labeled peptide was graded and loaded onto an Easy nLC HPLC system (Thermo Fisher, USA) for separation, and fragments were then analyzed by a Q-Exactive mass spectrometer (Thermo Fisher, USA). The raw peak data were further analyzed by Mascot 2.2 and Proteome Discoverer1.4 software. Bioinformatical Gene Ontology (GO) term enrichment analysis was conducted by the SRplot platform “http://www.bioinformatics.com.cn/ (accessed on 28 February 2024)” and the OmicShare tools “https://www.omicshare.com/tools (accessed on 28th February 2024)”.

### 2.15. Data Visualization and Statistical Analysis

The data were presented with mean ± SEM and visualized by Prism Graphpad 8.0. All experiment data were statistically analyzed by SPSS or Prism Graphpad 8.0. Briefly, the normal distribution of data was determined for each dataset by the Shapiro–Wilk W test; differences between the two groups were compared with Student’s *t*-test; and multiple group comparisons were tested by ANOVA with a Tukey’s post hoc test for normally distributed data with equal variance. Otherwise, either the Mann–Whitney U test or the Kruskal–Wallis test, followed by Dunn’s post hoc test, was used. Two-sided probability values < 0.05 were considered statistically significant.

## 3. Results

### 3.1. Knocking down HDAC6 Inhibits the Migration of HUVECs and Reduces ENG Expression

HDAC6 knockdown was induced in human umbilical vein endothelial cells (HUVECs) by RNA interference, and Western blot analysis showed that its expression was reduced by ~50% with both shRNA sequences; acetylated α-tubulin (a substrate of HDAC6) was significantly increased as a result (Figure 1G,H,J). HDAC6 knockdown decreased the migration ability of the HUVECs in the wound healing assay (Figure 1A,B). Transcriptomic and proteomic analyses were performed to map out genes with different expressions in HUVECs induced by HDAC6 knockdown. The transcriptomic data revealed that 5836 genes were dysregulated, 2773 were up-regulated, and 3063 were down-regulated (Figure 1C and Appendix A). At the protein level, 580 proteins were dysregulated, 256 were up-regulated, and 324 were down-regulated (Figure 1D and Appendix A). An integrated bioinformatics analysis of the datasets identified 273 genes that were differentially expressed at both the transcriptional and the protein levels (Appendix A). Subsequent Gene Ontology (GO) enrichment analyses revealed that these genes primarily contribute to angiogenesis-related biological processes (BP), such as response to wounding, regulation of cell migration, and regulation of cell motility, with 22 genes sharing these characteristics (Appendix A). Additionally, the GO enrichment for cellular components (CC) highlighted a significant association of these genes with cell–substrate junctions and focal adhesions, crucial for endothelial cell motility (Figure 1 and Appendix A). Combining the findings from both the BP and the CC GO enrichments, we identified 10 key proteins (Appendix A). A further review of the literature, along with our preliminary investigations performed on these genes, highlighted endoglin (ENG) as a critical factor in endothelial cell function. ENG’s down-regulation at both the mRNA and the protein levels in HUVECs, following HDAC6 knockdown, was confirmed through qPCR and Western blot analyses (Figure 1E–I).

### 3.2. BMP9 Promotes HUVECs Wound Healing and SMAD1/5/9 Phosphorylation in a Dose-Dependent Manner

To investigate whether ENG contributes to angiogenesis, we administered HUVECs with BMP9, which is an endogenous ligand of ENG. Wound healing results showed that BMP9 facilitated HUVEC migration at concentrations ranging from 1.25–20 ng/mL (Figure 2A,B). SMADs are the main signal transducers for receptors of the transforming growth factor beta (TGF-β) receptor superfamily, and both SMAD2/3 and SMAD1/5/9 complexes are implicated with angiogenesis; we assessed the phosphorylation of these SMAD complexes to determine the signaling pathways triggered by BMP9. Western blot analysis showed that phosphorylation of SMAD2/3 was unchanged with BMP9 treatment (Figure 2E,F). The phosphorylation level of SMAD1/5/9 gradually increased with the rising concentration of BMP9 treatment, reaching a peak at a BMP9 concentration of 5 ng/mL, which is 8.8 folds higher than that of the vehicle group (Figure 2C,D).

### 3.3. HDAC6 Mediates BMP9 Effects in Promoting Migration and Tube Formation of HUVECs

Considering that BMP9 can also bind to other receptors from the TGF-β superfamily, we performed various in vitro angiogenesis assays to explore whether the BMP9-induced angiogenesis was mediated by ENG and regulated by HDAC6. Our results showed that, similar to HDAC6 knockdown, ENG knockdown itself inhibited HUVEC wound healing and that the effects of BMP9 treatment (5 ng/mL) in promoting HUVEC wound healing were both blunted by ENG and HDAC6 knockdown (Figure 3A,B). The transwell and tube formation assays also confirmed that BMP9 treatment (5 ng/mL) significantly increased the migration and tube formation ability of HUVECs; HDAC6 and ENG knockdown not only reduced cell migration and impeded tube formation but also attenuated the stimulatory effects of BMP9 treatment (Figure 3C–G). Western blot results showed that HDAC6 knockdown reduced the increase in SMAD1/5/9 phosphorylation induced by BMP9 treatment (5 ng/mL), while knocking down ENG completely blunted this effect (Figure 3H,J). Notably, ENG protein expression in HUVECs significantly increased after BMP9 treatment (Figure 3H,I).

### 3.4. HDAC6 Regulates ENG Promoter Activity by Interacting with Transcription Factor SP1

Given the fact that HDAC6 knockdown inhibited ENG expression both at mRNA and at protein levels, we investigated whether HDAC6 affected its promoter activity. An HDAC6 knockout 293T cell line was established by the CRISPR-Cas9 gene editing system (Figure 4A), and a subsequent luciferase reporter assay demonstrated a decreased luciferase activity in the ENG promoter region (Figure 4B). Computational analyses, supported by previous studies [34,35], indicated that the transcription factor SP1 could bind to the ENG promoter sequence, thereby modulating ENG expression (Figure 4C). Co-immunoprecipitation (Co-IP) experiments confirmed a direct interaction between HDAC6 and SP1 (Figure 4D). In HUVECs, SP1 predominantly resides in the nucleus, with a smaller presence in the cytoplasm. Conversely, HDAC6 is mainly cytoplasmic and demonstrates partial co-localization with SP1, suggesting that their interaction largely occurs in the cytoplasm. Additionally, control group HUVECs display normal morphology, characterized by evident polarization and directional movement. However, in cells with HDAC6 knockdown, there is a noticeable decline in polarization, leading to a pancake-like cell appearance, and a concurrent decrease in SP1 expression is observed (Figure 5E,F).

### 3.5. HDAC6 CD2 Deacetylates SP1 and Regulates ENG-Mediated Angiogenesis

It is reported that post-translational modifications can alter the activity of SP1 [36,37,38]; we evaluated its acetylation level by immunoprecipitation of acetylated lysine. Our results showed that the acetylation of SP1 increased significantly in HUVECs infected with shHDAC6 lentivirus (Figure 5A,B). Since HDAC6 possesses two tandem catalytic domains with deacetylase activity and a ZNF domain that can interact with ubiquitinated proteins (Figure 5C), we explored which catalytic domain was responsible for deacetylating SP1 and whether other kinds of non-enzymatic activity of HDAC6 can also regulate ENG expression. We constructed an overexpression lentivirus of the HDAC6 CD1 inactive mutant H216A, CD2 inactive mutant H611A, and wild-type HDAC6 (Figure 5C). Our results demonstrated that overexpression of the wild-type HDAC6 and H216A mutants in HUVECs enhanced the effects of BMP9 treatment on wound healing, while overexpression of the CD2 inactive mutant H611A blunted the effects of BMP9 treatment on wound healing (Figure 5D,E). Further Western blot analysis revealed that overexpression of the wild-type HDAC6 and H216A mutants increased ENG expression and thereby increased SMAD1/5/9 phosphorylation after BMP9 treatment, while overexpression of the H611A mutant had no impact on ENG expression but did diminish the effects induced by BMP9 treatment (Figure 5F–H), indicating that HDAC6 CD2 enzymatic activity is required for HUVEC response to BMP9 treatment.

## 4. Discussion

Lysine acetylation modification is widely present in various proteins in the cytoplasm and nucleus. The physicochemical properties of acetylated proteins often differ from those of the original proteins, facilitating swift functional modulation without necessitating changes in protein expression levels. Thus, lysine acetylation has garnered significant attention in research due to its pivotal role in protein regulation.

Within this context, the HDAC family has become a focus of study due to its crucial role in regulating protein acetylation levels. It has emerged as a potential drug target protein in various diseases, including neurodegenerative diseases [39,40], cancer [41,42], and cardiovascular diseases [43,44]. Preliminary studies indicate that pan-HDAC inhibitor trichostatin A (TSA) blocked angiogenesis in vitro and in vivo [45]. However, the specific underlying mechanisms remain ambiguous. Subsequent investigations have revealed that both nuclear (HDAC1, 2, 3) and cytoplasmic (HDAC6) isoforms of HDAC can mediate angiogenesis [4,46,47,48], though their respective mechanisms diverge. Due to its unique structural composition, HDAC6 has complex intracellular functions, and since various extracellular and intracellular signaling molecules regulate angiogenesis [49], the mechanism by which HDAC6 mediates angiogenesis has not been fully elucidated.

The present study employs an integrated approach of transcriptomics and proteomics to investigate gene expression changes in vascular endothelial cells following HDAC6 knockdown. This approach led to the identification of ENG, a downstream protein regulated by HDAC6. ENG, a glycosylated membrane protein, is specifically expressed in endothelial cells and is part of the TGF-β receptor superfamily [50]. It predominantly binds to the ligands BMP9 and BMP10 [51,52]. ENG is known to play a crucial role in maintaining vascular integrity [50,53]. Homozygous mutations in mice result in embryonic lethality, and certain point mutations in human ENG are linked to hereditary hemorrhagic telangiectasia (HHT) [50]. However, the relationship between ENG, its ligand BMP9, and angiogenesis has been ambiguous, with conflicting reports regarding BMP9′s role in angiogenesis [54,55,56,57,58]. In this study, the effects of BMP9 on angiogenesis were assessed using various models, including scratch wound healing, transwell migration, and tube formation assays in HUVECs. The results indicated that BMP9 treatment (5 ng/mL) significantly promotes endothelial cell scratch wound healing, transmembrane migration, and tube formation, thereby affirming BMP9’s role in angiogenesis. Additionally, the study revealed that knockdown of HDAC6 and ENG not only reduced the angiogenic capabilities of HUVECs but also blunted the effects of BMP9 treatment. Western blot analysis showed a significant increase in SMAD1/5/9 phosphorylation following BMP9 treatment, which diminished with HDAC6 knockdown, accompanied by a decrease in ENG expression. This pattern mirrors the effects observed with direct ENG interference, suggesting that HDAC6 modulates BMP9-induced angiogenesis through ENG expression regulation. Intriguingly, BMP9 treatment was found to substantially elevate ENG protein levels, hypothesized to be a compensatory cellular response to prolonged ligand–receptor binding leading to receptor desensitization.

Given that HDAC6 knockdown suppressed ENG expression at both the mRNA and the protein levels, it is proposed that HDAC6 influences ENG transcription through modulation of promoter activity. This was proved by luciferase reporter assays in HDAC6 knockout 293T cells, which showed a significant decrease in ENG promoter activity. Computational analysis of transcription factor binding sites identified transcription factor SP1 as a key regulator of ENG promoter activity. Immunoprecipitation experiments confirmed a direct interaction between HDAC6 and SP1. Previous studies suggested that SP1 activity is directly regulated by its acetylation level, with higher acetylation correlating to lower transcriptional activity [36,37,38]. This was observed in HDAC6 knockdown HUVECs, where an increase in SP1 acetylation was noted. Considering HDAC6’s structural complexity, the study further explored its impact on ENG expression and BMP9 treatment response by overexpressing mutant HDAC6 proteins with CD1 inactive (H216A) and CD2 inactive (H611A). Results showed that overexpression of both the CD1 mutant H216A and wild-type HDAC6 enhanced ENG expression and augmented BMP9′s effects, whereas the CD2 mutant H611A did not produce similar effects, suggesting a possible compensatory decrease in endogenous HDAC6 levels. This finding confirms HDAC6’s role in mediating SP1 acetylation regulation through its CD2 enzymatic activity, thereby influencing SP1 activity and ultimately regulating ENG expression.

Despite the progress made, our study faced several limitations that warrant further investigation. The integrated analysis of transcriptomic and proteomic data pinpointed several other proteins that were down-regulated in tandem with HDAC6 knockdown and were closely related to endothelial cell functions, including integrin β1, integrin β3, nitric oxide synthase (NOS), etc. The mechanisms through which HDAC6 regulates these proteins, as well as their specific contributions to the angiogenic process, remain to be fully elucidated. Furthermore, additional research is required to precisely determine how HDAC6 influences SP1 acetylation at specific sites and how this modulation affects SP1’s interactions with other transcription factors and DNA. Moreover, considering the variable effects of BMP9 on angiogenesis in different endothelial cell types [58,59], it is important to investigate the potential competitive interaction among other types of TGF-β receptors, such as TGF-β receptor 2 (TGFBR2), serine/threonine-protein kinase receptor R3 (ALK1), and ENG in BMP9 binding. Then, the differential responses of endothelial cells to BMP9 might be the results of varying expression levels of these TGF-β receptor subtypes.

## 5. Conclusions

In conclusion, this study demonstrates that HDAC6, through its CD2 catalytic domain, modulates the acetylation level of transcription factor SP1, regulating ENG expression and playing a significant role in BMP9-induced angiogenesis.

## Figures and Tables

**Figure 1 cells-13-00490-f001:**
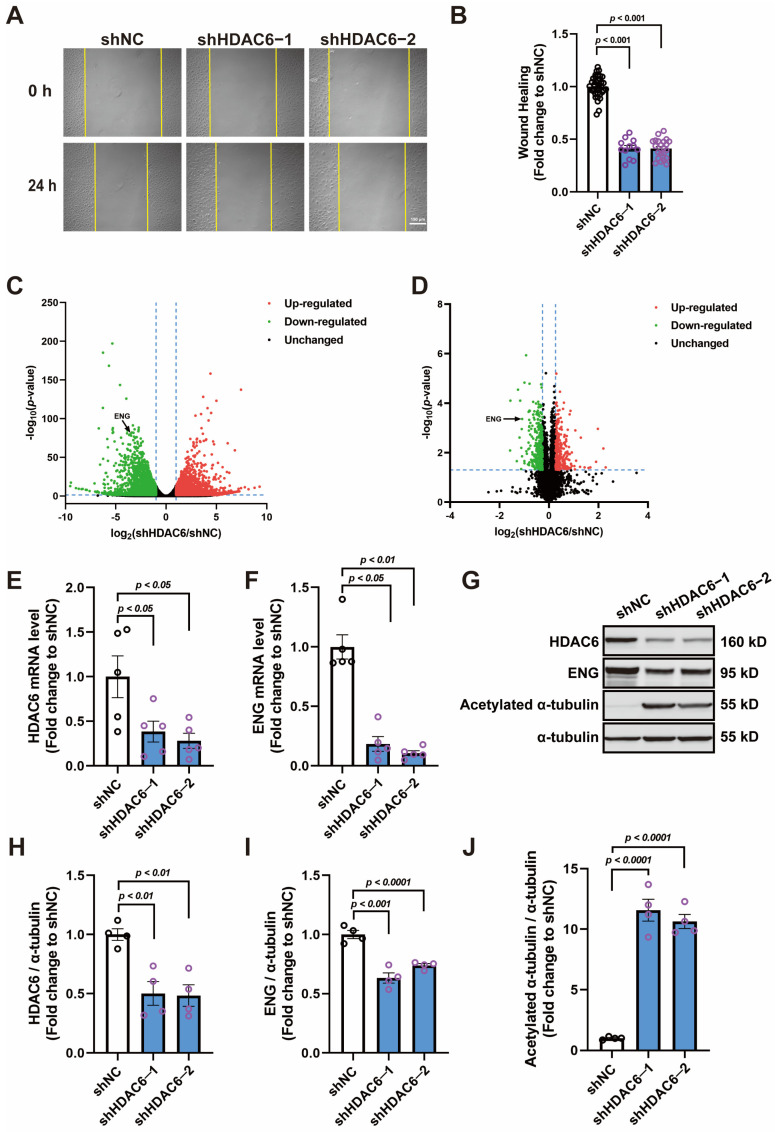
Knocking down HDAC6 inhibits the migration of HUVECs and reduces ENG expression. (**A**,**B**) Wound healing of HUVECs infected with shNC (scramble shRNA control), shHDAC6−1, or shHDAC6−2 lentivirus; *n* = 12–36; bar = 150 μm. (**C**) Volcano plot of transcriptomic analysis between the control (shNC) and HDAC6 knockdown (shHDAC6) HUVECs; *n* = 3. (**D**) Volcano plot of proteomic analysis between the control (shNC) and HDAC6 knockdown (shHDAC6) HUVECs; *n* = 3. (**E**,**F**) mRNA expression of HDAC6 and ENG in HUVECs infected with shNC, shHDAC6−1, or shHDAC6−2 lentivirus; *n* = 5. (**G**–**J**) Protein expression of HDAC6, ENG, and acetylation of α-tubulin in HUVECs infected with shNC, shHDAC6−1, or shHDAC6−2 lentivirus; *n* = 4.

**Figure 2 cells-13-00490-f002:**
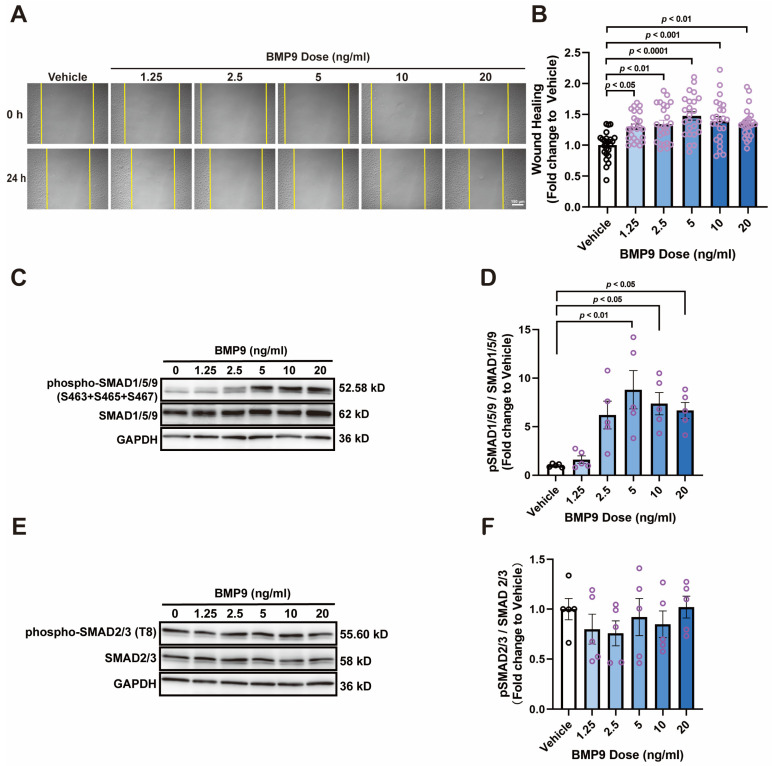
BMP9 promotes HUVEC wound healing and SMAD1/5/9 phosphorylation in a dose-dependent manner. (**A**,**B**) Effects of BMP9 treatment (1.25–20 ng/mL) on wound healing of HUVECs; *n* = 24; bar = 150 μm. (**C**,**D**) Phosphorylation of SMAD1/5/9 (S463 + S465 + S467) in HUVECs treated with BMP9 (1.25–20 ng/mL); *n* = 5. (**E**,**F**) Phosphorylation of SMAD2/3 (T8) in HUVECs treated with BMP9 (1.25–20 ng/mL); *n* = 5.

**Figure 3 cells-13-00490-f003:**
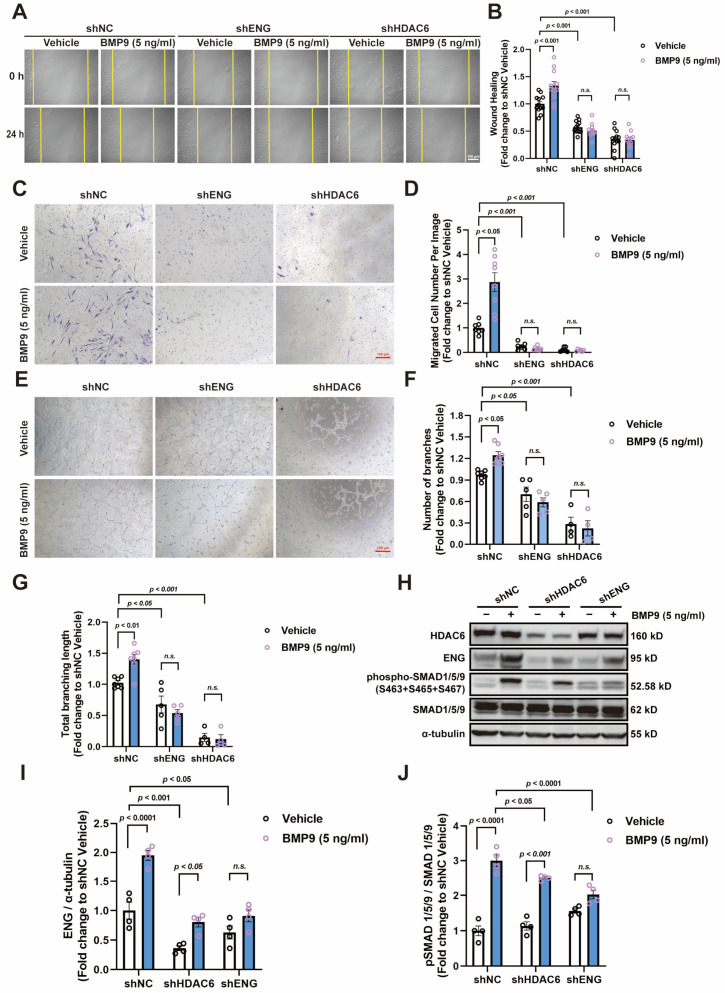
HDAC6 mediates BMP9 effects in promoting migration and tube formation of HUVECs. (**A**,**B**) Effects of BMP9 treatment (5 ng/mL) on wound healing in HUVECs with or without knockdown of HDAC6 and ENG; *n* = 12; bar = 150 μm. (**C**,**D**) Effects of BMP9 treatment (5 ng/mL) on the migration of HUVECs in a transwell model with or without knockdown of HDAC6 and ENG; bar = 150 μm; *n* = 6–8. (**E**–**G**) Effects of BMP9 treatment (5 ng/mL) on the tube formation of HUVECs with or without knockdown of HDAC6 and ENG, cultured on a matrigel matrix; bar = 150 μm; *n* = 4–7. (**H**–**J**) Effects of BMP9 treatment (5 ng/mL) on ENG expression and phosphorylation of SMAD1/5/9 in HUVECs with or without knockdown of HDAC6 and ENG; *n* = 4. n.s., not significant.

**Figure 4 cells-13-00490-f004:**
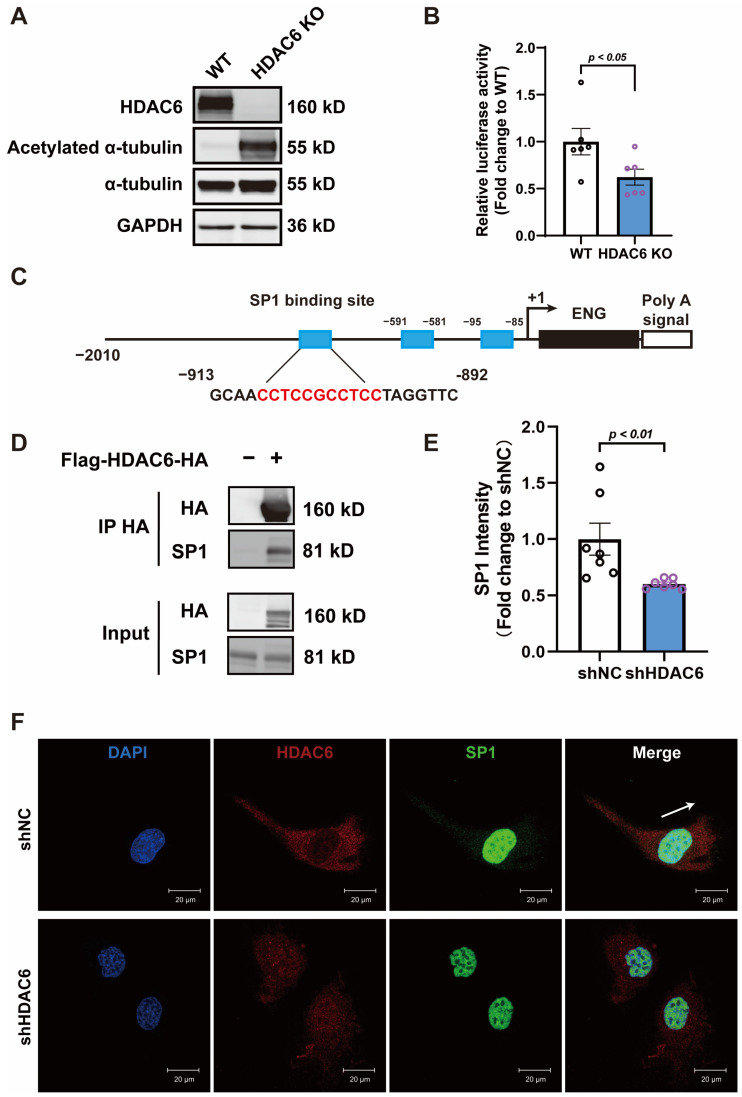
HDAC6 regulates ENG promoter activity by interacting with transcription factor SP1. (**A**) Representative Western blot image of HDAC6 expression and acetylation of α-tubulin in wild-type (WT) 293T and HDAC6 knockout 293T cells; *n* = 4. (**B**) Effect of HDAC6 knockout on the promoter activity of ENG; *n* = 6. (**C**) Transcription factor SP1 binding sites on ENG promoter predicted by JASPAR database; cyan square demonstrates proximal binding sites; red letters indicate the actual binding sequences. (**D**) Co-immunoprecipitation showed a direct interaction between the HA-tagged HDAC6 and SP1 in 293T cells; *n* = 3. (**E**,**F**) Representative photomicrographs showing the expression and distribution of HDAC6 and SP1 in HUVECs with or without HDAC6 knockdown; white arrow indicates the direction of cell movement; *n* = 7; bar = 20 μm.

**Figure 5 cells-13-00490-f005:**
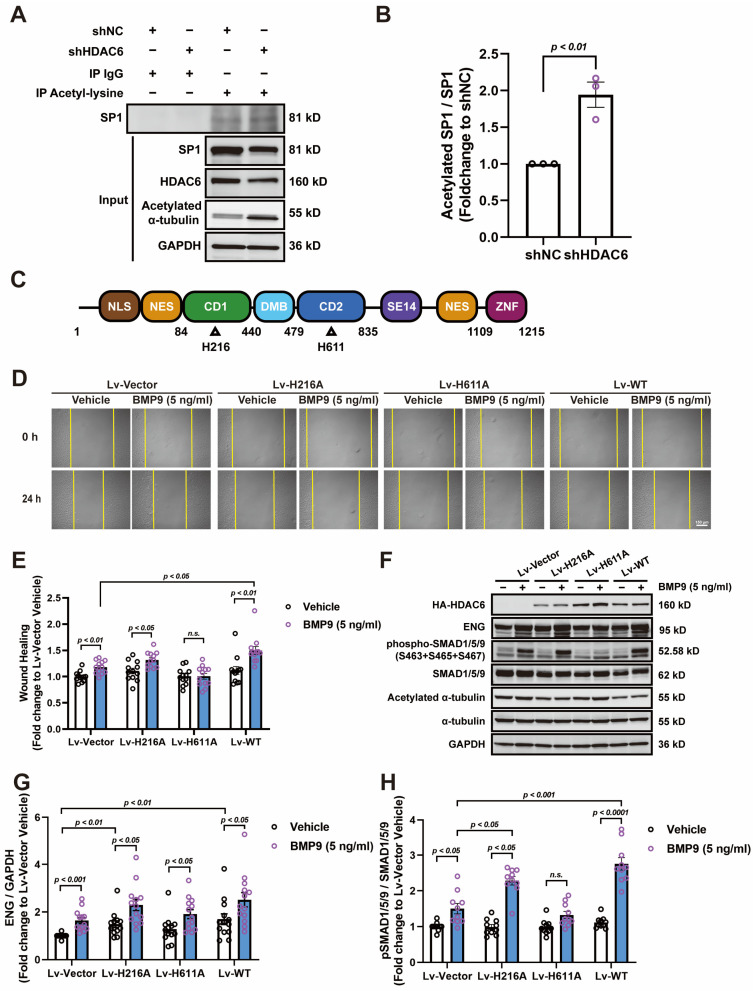
HDAC6 CD2 deacetylates SP1 and regulates ENG-mediated angiogenesis. (**A**,**B**) Immunoprecipitation of acetylated lysine in HUVECs with or without HDAC6 knockdown; GAPDH was used as a loading control *n* = 3. (**C**) Schematic structure of human HDAC6 protein: NLS—nuclear localization signal, NES—nuclear export signal, CD1—catalytic domain 1, DMB—dynein motor binding position, CD2—catalytic domain 2, SE14—cytoplasmic anchoring domain with serine-glutamate tetradecapeptide repeat, and ZNF—zinc finger motif. The enzymatic active histines targeted for inactive mutation were indicated below each catalytic domain. (**D**,**E**) Effects of BMP9 treatment (5 ng/mL) on wound healing in HUVECs with or without overexpression of HDAC6 mutants; H216A—CD1 inactive mutant, H611A—CD2 inactive mutant, and WT—wild-type human HDAC6; *n* = 12; bar =150 μm. (**F**–**H**) Effects of BMP9 treatment (5 ng/mL) on protein expression of ENG, phosphorylation of SMAD1/5/9, and acetylation of α-tubulin in HUVECs with or without overexpression of HDAC6 mutants; *n* = 10–13.

## Data Availability

All supporting data and materials are available online.

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
