# Peer review of "HDAC6 Enhances Endoglin Expression through Deacetylation of Transcription Factor SP1, Potentiating BMP9-Induced Angiogenesis"

_cells, 2024, doi:10.3390/cells13060490_

Round 1

Reviewer 1 Report

Comments and Suggestions for Authors

I think that the topic is of interest and owrth to be published; I have several minor issues? 

At the end of the introduction - There is no question or aim mentioned - instead of this they present a result or a conclusion - this is not the right plkace for this? It is better to mention a question or the aim of the study!! What are the limitations of this study? These are not really mentioned and discussed - please do so! What is missing, what is needed? Please state and discuss!

Reviewer 2 Report

Comments and Suggestions for Authors

The researchers investigated the role of Histone deacetylase 6 (HDAC6) in angiogenesis, particularly its impact on the acetylation of non-histone proteins. By conducting transcriptomic and proteomic analyses on vascular endothelial cells with HDAC6 knockdown, the study identified endoglin (ENG) as a key downstream protein regulated by HDAC6. Mechanistically, HDAC6 was found to modulate ENG transcription by influencing promoter activity, resulting in increased acetylation of the transcription factor SP1 and altering its transcriptional activity. This work has the following points to be addressed before it can be published:

  1. Correlating both the transcriptomics and proteomics data, there are over 300 matching genes/proteins that were down regulated. Quite a few of them are integrin proteins, thrombomodulin, VWF, nitric oxide synthase, EDN1 etc.,which are also associated with endothelial cell function. 

Why was endoglin specifically picked out for this study? What “integrated analysis” was done to have facilitated the authors to focus on endoglin? The author needs to elaborate more on this point.

2. The legends for the bar size of all the wound healing assay figures are missing. Although it’s detailed in the corresponding figure legends, it would be much clearer to put the legend right next to the bar.

Reviewer 3 Report

Comments and Suggestions for Authors

The manuscript by Sun and colleagues studied the contribution of HDAC6 on angiogenesis, in vitro, using HUVEC and performing cell migration and tube formation. They showed that downregulation of HDAC6 in HUVEC decreased the expression of END by reducing deacetylation of the transcription factor SP1. Since END is a receptor for BMP-9, they showed that downregulation of HDAC6 significantly influenced the pro-angiogenic functions of HUVECs mediated by BMP-9. 

The framework of the molecular pathways analysed is well organised and the results contribute to a better understanding of the intricate process of angiogenesis.

Concerns

HUVEC cells are used for how many passages? From which passage after downregulation of HDAC6? Please report these data in the main manuscript.

It is rather curious that the authors used a stable knockdown of HDAC6 by infection with shRNA in HUVECs. Why did they choose this technology?

Lane 204: Please exchange 580 genes for 580 proteins.
